# Cortical Oxygenation Changes during Gastric Tube Feeding in Moderate- and Late-Preterm Babies: A NIRS Study

**DOI:** 10.3390/nu13020350

**Published:** 2021-01-25

**Authors:** Mariana Muelbert, Tanith Alexander, Chris Pook, Yannan Jiang, Jane Elizabeth Harding, Frank Harry Bloomfield

**Affiliations:** 1Liggins Institute, University of Auckland, 1142 Auckland, New Zealand; m.muelbert@auckland.ac.nz (M.M.); t.alexander@auckland.ac.nz (T.A.); chris.pook@auckland.ac.nz (C.P.); y.jiang@auckland.ac.nz (Y.J.); j.harding@auckland.ac.nz (J.E.H.); 2Neonatal Unit, Kidz First, Middlemore Hospital, 2025 Auckland, New Zealand

**Keywords:** cortical activation, tube feeding, smell, taste, milk, sensory stimulation, near-infrared spectroscopy, moderate and late preterm, nutrition

## Abstract

Smell and taste of food can trigger physiological responses facilitating digestion and metabolism of nutrients. Controlled experimental studies in preterm babies have demonstrated that smell activates the orbitofrontal cortex (OFC) but none have investigated the effect of taste stimulation. Using cotside Near-Infrared Spectroscopy (NIRS), we measured changes in OFC cerebral oxygenation in response to gastric tube feeds five and 10 days after birth in 53 assessments of 35 moderate- to late-preterm babies enrolled in a randomized trial. Babies were randomly assigned to receive smell and taste of milk before gastric tube feeds (intervention group, *n* = 16) or no exposure (control group, *n* = 19). The majority of babies were born at 33 weeks of gestation (range 32–34) and 69% were boys. No differences in OFC cerebral oxygenation were observed between control and intervention groups. Gastric tube feeds induced activation of the OFC (*p* < 0.05), but sensory stimulation alone with smell and taste did not. Boys, but not girls, showed activation of the OFC following exposure to smell of milk (*p* = 0.01). The clinical impact of sensory stimulation prior to tube feeds on nutrition of preterm babies, as well as the impact of environmental inputs on cortical activation, remains to be determined.

## 1. Introduction

Smell and taste are intimately connected to nutrition and behavior, directly influencing food preferences [1,2]. Sensory cues originating from smell and taste of food can trigger physiological responses facilitating digestion and metabolism of ingested nutrients [3]. The integration of taste and smell information arising from the peripheral sensory organs is believed to occur in the orbitofrontal cortex (OFC) [4,5,6,7]. The OFC integrates information from the five sensory modalities (visual, auditory, somatosensory, gustatory, and olfactory), forming a complex network with other brain areas, such as the insula, amygdala, and piriform cortex [8].

Neuronal activity induces an increase in cerebral blood flow to supply the oxygen required to meet cellular metabolic demand [9]. During sensory stimulation there is a rise in oxygen delivery to the brain cells, which leads to an increase in concentration of oxygenated hemoglobin in the cerebral cortex, including the OFC [8,10]. Changes in hemoglobin oxygen content provide information about oxygen delivery and utilization by neuronal cells, coupled to cerebral activity [11], and can be measured non-invasively by near-infrared spectroscopy (NIRS) [10]. Oxygenated (O_2_Hb) and deoxygenated (HHb) hemoglobin absorb near-infrared light at different wavelengths [12], which can be detected by NIRS, thereby allowing estimation of changes in the oxygen content of hemoglobin associated with cortical activity [11,13].

Changes in cerebral oxygenation in the OFC have been associated with neuronal activation of the olfactory cortex [14,15,16,17,18] and can be detected by NIRS following exposure to a variety of odors in adults [19] and term and preterm babies [14,15,16,17,18], including the odor of the maternal breast [18], nosocomial odors [14,17], and odor of breastmilk and formula [16]. Similarly, oxygenation changes in the prefrontal cortex measured by NIRS have been associated with processing of taste information in adults [4,20].

Provision of smell of breastmilk in tube-fed preterm infants has been linked to increased sucking behavior and milk ingestion [21,22], and possibly may lead to more rapid attainment of full oral feeds and reduced length of hospital stay [23], although the quality of evidence is very low. Similarly, oral administration of small amounts of colostrum has been associated with faster attainment of full tube feeds but, again, evidence is of very low quality [24].

The studies linking smell to potential activation of the OFC in preterm babies have been conducted in carefully controlled settings and there are no studies investigating cortical activation with taste stimulation in preterm infants. It is, therefore, not clear whether activation of the OFC can be detected in preterm babies in the usual setting of tube feeds in a newborn nursery or whether the administration of milk through a gastric feeding tube itself can lead to increases in oxygenation in the OFC. Thus, our study aimed to describe cerebral oxygenation changes in the OFC measured by NIRS in moderate-late preterm babies (birth between 32 and 36 weeks of gestation) following exposure to smell and taste of milk and during tube feeds.

## 2. Materials and Methods

### 2.1. Study Population

This was a cohort study nested within a multicenter, factorial, randomized controlled trial, the DIAMOND Trial (DIfferent Approaches to MOderate & late preterm Nutrition: Determinants of feed tolerance, body composition and development) [25], and was approved by the national Health and Disability Ethics Committee (HDEC 16/NTA/90). Written informed consent was obtained from parents or caregivers. Briefly, the DIAMOND trial is investigating the impact of different nutritional approaches on feed tolerance, body composition, and neurodevelopment in moderate-late preterm (MLP) babies. The three main factors under investigation are: (1) provision of intravenous nutrition with parenteral nutrition compared to intravenous dextrose, (2) provision of enteral nutrition with exclusive maternal breastmilk compared to milk supplementation with infant formula or donor breastmilk, and (3) exposure to smell and taste of milk before all tube feeds compared to no smell and taste of milk prior to tube feeds. Eligible babies were born between 32^+0^ and 35^+6^ weeks of gestation, admitted to one of five neonatal intensive care units (NICU) in New Zealand, had an intravenous canula sited for clinical reasons, and their mother intended to breastfeed. Babies with congenital abnormality or for whom a particular mode of nutrition was clinically indicated were not eligible.

The current study included a subsample of participants in the DIAMOND trail who were born before 35 weeks of gestation, exclusively tube fed and admitted to the NICU at Auckland City Hospital (ACH), and whose parents/guardians gave informed consent for this additional NIRS assessment. We did not include babies born ≥35 weeks of gestation as these babies often initiate oral feeding attempts within the first postnatal days. Thus, the sample size was determined by the number of consented babies <35 weeks of gestation at one site enrolling babies into the DIAMOND trial.

### 2.2. Assessment Procedure

NIRS assessments were performed on day 5 (±2 days) and day 10 (±2 days) after birth, during feeds where there was no sucking attempt at the breast prior to the tube feed. The assessment sequence was adapted from previous studies with newborn/preterm babies [16,18], who were provided smell exposure for 30 s. For the intervention group, the assessment consisted of a 5-min baseline reading period (P0), a 1-min period of exposure to smell (P1), a 2-min interval, then a 1-min period of exposure to taste (P2), another 2-min interval, then a period of tube feeding of variable duration (P3). Babies in the control group were monitored for the equivalent time; however, no placebo or sham stimulation was delivered. The total assessment duration for both groups consisted of 11 min before the feed plus the duration of the feed, which varied according to the volume of feed (Figure 1). As no previous study had investigated cerebral oxygenation changes following exposure to taste in babies, we undertook some pilot assessments, which determined that one minute of sensory exposure was feasible and well tolerated by babies. For consistency, equal duration of sensory stimulation was used for both smell and taste exposure.

### 2.3. Sensory Stimulation

Smell and taste stimulation were provided using the milk fed to the infant at the time of the assessment (either breastmilk, fortified breastmilk, or infant formula). Smell stimulation was provided during P1 by placing a gauze containing approximately 0.2 mL of milk close to the baby’s nose and moving this slowly from one nostril to the other at a distance of approximately 1–2 cm for 1 min. For taste exposure during P2, a syringe containing 0.2 mL of milk was used to place drops of milk into the baby’s mouth for 1 min. Gastric tube feeding (P3) was administered according to local NICU guidelines.

### 2.4. Cerebral Oxygenation Monitoring

Cerebral oxygenation levels were monitored at the bedside using a double-channel NIRS device (NIRO-200, Hamamatsu, Japan) at wavelength of 775–850 nm. Concentrations of oxygenated hemoglobin (O_2_Hb) were recorded every second and stored in a dedicated personal computer. Onset and completion of P1, P2, and P3 were marked on the NIRS device. Emitter and receiver optodes were placed bilaterally 2 cm above the midpoint of the line connecting the external angle of the eye to the homolateral tragus, reflecting placement anterior to T3/T4 and anterior to F7/F8 in the international 10–20 system for electrode placement [26], overlying the orbitofrontal cortex (Figure 2). The differential pathlength was set to 3.85 cm and optical pathlength to 11.5 cm. The transmitter and receiver optodes were placed 3 cm apart in a purpose-designed holder attached to the forehead with double-sided adhesive tape, covered by a bandage. Babies were swaddled and incubators covered consistent with standard practice in this nursery. Room light and noise were diminished as much as possible to minimize interference; however, the room environment was kept as close as possible to daily routines to mimic the real-life scenario.

### 2.5. Data Processing and Statistical Analysis

In order to minimize slow oscillations and artefacts common to NIRS recordings, data smoothing was performed using a 15-s moving average. Data points that were more than 2 standard deviations from the mean were considered artefacts and were excluded. Baseline O_2_Hb concentration for each baby was defined as the mean of a 30-s stable period within P0. The maximum change in O_2_Hb concentration (mmol/L) from baseline was then calculated for P1, P2, and P3 separately for left and right hemispheres. To explore the effect of exposure to smell and taste of milk and tube feeding on change in O_2_Hb concentrations, we used linear mixed model regression analysis with repeat measurements clustered within each baby and assumed an unstructured covariance matrix between the measurements (P1, P2, and P3). We tested the interaction between O_2_Hb changes, group and optode location (Model 1—left *versus* right), postnatal day at assessment (Model 2—day 5 *versus* day 10), and sex (Model 3—boys *versus* girls). Model-adjusted estimates for each group at each period and assessment were estimated using fixed effects models, adjusted for sex, gestation, and baseline oxygenation. All variables were tested for normality before analysis. Statistical analyses were performed with SAS version 9.4 (SAS Institute Inc., Cary, NC, USA).

## 3. Results

### 3.1. Study Population

Between June 2018 and March 2020, 62 moderate-late preterm babies admitted to NICU at Auckland City Hospital were enrolled in the DIAMOND trial and 50 parents/caregivers consented to NIRS assessments. Thirty-five of these underwent the first assessment at a median postnatal age of 4 (range 3–6) days and 18 underwent the second assessment at a median age of 10 (range 8–12) days (Figure 3). Median gestational age at birth was 33 (range 32–34) weeks, 69% of babies were boys, and the majority of babies received breastmilk as their feed during the assessments. Baseline characteristics did not differ between control and intervention groups (Table 1).

### 3.2. Model 1: The Effect of Laterality

This model included data from the first assessment only and data from three participants were excluded for signal artefacts that rendered the data of too poor quality for analysis, resulting in 32 assessments (18 in the control and 14 in the intervention group) included. There was no difference in lateral cortical activation on either side between babies in the control and intervention groups for any period (*F_(2,59)_* = 0.88, *p* = 0.4) (Table 2). Babies in the intervention group showed significant increase in O_2_Hb concentrations from baseline for each of the three periods on the right side and for P2 (taste) on the left side of the brain (*p* < 0.05). Babies in the control group showed a significant increase in O_2_Hb concentrations from baseline on the left side for P1 and bilaterally for P3 (*p* < 0.05). They also showed a significant increase from P2 to in P3 on the left side (*p =* 0.01) (Figure 4A).

### 3.3. Model 2: The Effect of Postnatal Day

As there were no differences in O_2_Hb concentrations between the right and left sides, measurements from both sides were combined in order to explore the effect of postnatal age at the assessment. This model included data from first and second assessments (*n* = 53) and excluded data from first assessment from three participants were imputed by maximum likelihood. Overall, there was no significant interaction between period, study group, and postnatal day (*F*_(2,95)_ = 0.41, *p* = 0.7); however, there was a significant interaction between postnatal day and study period (*F*_(2,95)_ = 5.29, *p* = 0.006) (Table 2). In both study groups, O_2_Hb concentrations measured during P2 were significantly higher in the first than the second assessment (*p* < 0.001). Further, O_2_Hb concentrations in the control group were significantly higher during P3 (feeding) than in P1 and P2 (*p* < 0.05) in both assessments. Similarly, in the intervention group O_2_Hb concentrations in the second assessment were significantly higher during P3 than during P2 (*p* < 0.001) (Figure 4B).

### 3.4. Model 3: The Effect of Sex

This model included the first assessment only (*n* = 32) as only one girl in each group was assessed twice. There was a significant interaction between period, study group, and sex (*F*_(2,95)_ = 4.95, *p* = 0.01). In girls in the intervention group O_2_Hb concentration during P2 and P3 (taste and feed) was higher than baseline and P1 (smell) (Table 2). In contrast, in girls in the control group O_2_Hb concentration was higher than baseline during P1 (*p* = 0.01) and also significantly higher than the intervention group (*p* = 0.03).

Conversely, in boys in both groups O_2_Hb concentrations were significantly increased from baseline concentration in all periods (*p* < 0.01) but there were no significant differences among periods or between groups.

Finally, when comparing boys and girls within groups, we found that only boys in the intervention group had an increase in O_2_Hb concentrations during P1 (smell of milk) and this was significantly higher than in girls exposed to smell of milk (*p* = 0.002) (Figure 4C).

## 4. Discussion

### 4.1. Main Findings

In contrast to our hypothesis and to previous studies exposing babies to smell in highly controlled settings [15,16,17,18], we did not find that exposure to smell activated the OFC in moderate-late preterm babies. However, changes in O_2_Hb concentration during exposure to smell of milk were significantly higher in boys than in girls, whereas girls only showed an increase of O_2_Hb concentrations during exposure to taste. Further, we found that in the clinical, rather than experimental, setting, gastric tube feeds resulted in the greatest change in OFC oxygenation.

### 4.2. The Effect of Side

We found similar O_2_Hb concentration changes in both left and right OFC with no clear evidence of laterality of response to sensory stimulation. Previous studies in well-controlled settings [16,17,18] have shown that newborn babies are aware of their olfactory environment and are capable of discriminating between different smells, but the evidence for laterality in the response is not consistent. For example, in full-term babies exposed to smell of colostrum and vanilla, increases in oxygenation were detected in the left OFC [15], whereas a study including both term and preterm babies reported bilateral increases in OFC O_2_Hb concentrations that were greater with breastmilk than with formula [16]. In contrast, exposure to unpleasant smells (such as detergent and adhesive remover) resulted in decreased OFC oxygenation that was more marked on the right than the left [14]. Conversely, in studies including both preterm and term babies, bilateral increase in oxygenation in the OFC were observed following exposure to unpleasant smells (hand sanitiser and adhesive remover), with preterm babies showing greater cortical activation at term-corrected age compared to in the early postnatal period [17]. There is some evidence that laterality may be affected by maturity, with one study reporting that exposure to the smell of the maternal breast elicited no cortical activation in very preterm babies, only right cortical activation in late preterm babies, but bilateral cortical activation in full-term babies, leading the authors to suggest that odor processing at a cortical level may be present in neonates born from 32 weeks of gestation onwards [18]. As our cohort was limited to moderate- and late-preterm babies, with most falling in the moderate-preterm category, we were unable to assess the impact of the degree of prematurity in cortical processing of smell.

### 4.3. The Effect of Postnatal Day

Lower O_2_Hb concentrations in response to milk placed in the mouth (P2) were observed during the second assessment compared to the first. It is possible that babies developed habituation to taste of milk with increasing exposure through breastfeeding attempts/trials, resulting in weaker activation of the OFC. Habituation is often described in adults as a reduction in physiological and behavioral responses following repeated eating episodes [27]; however, it is yet to be determined if this occurs in preterm babies. Decreasing O_2_Hb concentrations in response to smell of colostrum with increasing postnatal age has been reported previously [15], potentially indicating that older babies more familiarized with smell of milk through previous feeding experiences became less sensitive to smell of milk as feeding frequency increased [15]. However, a study investigating whether newborn babies (preterm and term) could discriminate between the smell of breastmilk and formula, in which babies were grouped according to their feeding experience (exclusive breastmilk versus mixed feeding), did not find any interaction between previous feeding experience and OFC oxygenation [16]. In our study, the majority of babies were fed breastmilk (with or without multi-component bovine-based fortifier) during the assessment and, thus, we were unable to further clarify the effect of milk type on OFC oxygenation.

### 4.4. The Effect of Sex

We found contrasting patterns of oxygenation in the OFC for boys and girls. Boys showed a significant increase in O_2_Hb concentration when exposed to smell of milk and also during the feeding period, whereas girls showed no changes in O_2_Hb concentration in response to smell of milk, but a steep increase in O_2_Hb concentration in response to taste of milk. This is consistent with a previous report of difference between boys and girls in response to smell of the maternal breast, where boys showed bilateral activation of the olfactory cortex but girls showed activation only of the left olfactory cortex [18]. In general, women are reported to be more sensitive than men to a variety of odors [28], possibly due to differences in circulating neuroendocrine hormones [28]. The possible differences in OFC oxygenation between preterm girls and boys, and whether this has any clinical relevance, requires further investigation.

### 4.5. OFC Activation without Sensory Stimulation

In contrast to our hypothesis, the control group showed an increase in O_2_Hb concentration during P1 and P2 despite receiving no sensory stimulation. Although Frie et al. (2020) reported bilateral activation of the frontal cortex when late-preterm babies were exposed to a control stimulus, consisting of a clean cloth washed with odorless detergent [18], in our study no stimulus at all was provided.

However, in our study the majority of babies were awake during the assessments and it seems possible that infants in the control group were responding to smells or other sensory input originating from their surroundings. The NICU environment provides numerous sensory inputs, including handling, light, noise, and exposure to nosocomial odors, all of which may impact the development of sensory systems of the immature brain [29,30]. We designed our study to be in the normal NICU environment, as we wished to determine whether smell and taste have any clinically relevant impact, so we cannot exclude the possibility that other sensory inputs were inducing cortical activation. Since our focus was on activation of the OFC, the optode pairs were placed only over this region, but the responses observed in the control group could reflect more general cortical activation and not activation of the OFC alone. Future studies are needed to confirm potential cortical activation related to NICU environment stimuli.

### 4.6. The Effect of Feeding

Research into the potential role of smell exposure in preterm babies is predicated upon the assumption that tube-fed babies miss out on a potentially beneficial sensory stimuli that may promote feed tolerance and digestion through activation of the cephalic phase response [23]. Our unexpected finding that the greatest increase in O_2_Hb concentration in the OFC occurred in response to tube feeds raises the possibility that this mode of feeding does, in fact, activate olfactory and/or gustatory receptors. This may occur through various routes. For example, olfactory receptors have been found throughout the gastrointestinal tract and in organs that assist digestion and nutrient metabolism, such as the pancreas and liver, and may possibly be stimulated during feeding, mediating the secretion of digestive hormones [31,32,33]. Similarly, taste receptors are expressed throughout the gastrointestinal epithelium and are involved in regulation of food intake, digestion, and the secretion of the digestive hormones such as ghrelin, leptin, glucagon, and cholecystokinin [33]. Whether these receptors located outside of the oronasopharynx activate the OFC is unknown. Alternatively, it is possible that small quantities of milk, or the volatile compounds within milk responsible for smell, reflux from the stomach into the upper gastrointestinal tract, activating olfactory and gustatory receptors and the OFC. The presence of a feeding tube passing through the gastro-esophageal junction increases the likelihood of such reflux by maintaining patency of that junction [34]. Indeed, gastroesophageal reflux is a normal physiological phenomenon in most healthy babies and can occur in every feed with or without vomiting [35].

### 4.7. Strengths and Limitations

We nested this experimental cohort study within the DIAMOND trial to take advantage of the random allocation of babies to experimental and control groups. This allowed us to test the effect of exposing tube-fed babies to smell and taste of milk before and during feeds compared to no sensory stimulation with tube feeds. We also aimed to assess babies at the bedside in an environment similar as possible to the usual clinical setting, and most babies were awake during the assessment, which may have influenced the baby’s level of awareness of the surroundings and affected the measurement of the outcome. In contrast, no previous studies investigated changes in O_2_Hb concentrations in the context of feeding, and all used a carefully controlled environment and assessed babies while asleep.

Given the study design, it was not possible to blind the assessor to the intervention and no control stimulation was used, which may be considered limitations. Our study also had a small sample size, limited by the number of trial participants whose parents/caregivers consented for the assessment and the fact that NIRS was available at only one recruiting site, and the unequal distribution of moderate- and late-preterm infants did not permit us to assess the effect of degree of prematurity on activation of OFC. Similarly, most babies received breastmilk or fortified breastmilk during NIRS assessments and, therefore, we did not have sufficient numbers to analyze the effect of type of feeding on oxygenation changes in the OFC. Additionally, we used a double-channel NIRS device with only two pairs of emitter–receiver optodes, which may have less spatial resolution and less sensitivity to detect changes in cortical oxygenation [13] compared to multi-channel devices with a larger number of emitter–receiver optodes [17,18]. Nevertheless, our findings are consistent with other studies using the same device [14,15,16].

## 5. Conclusions

We showed that changes in cerebral oxygenation in the OFC can be detected in moderate- and late-preterm infants before and during tube feeding. The greatest increase in O_2_Hb concentration in the OFC occurred in response to tube feeds, suggesting that gustatory and olfactory receptors may be activated in this feeding mode. Boys but not girls showed activation of the OFC following exposure to smell of milk. Early-life sensory exposures are critical for the development of smell and taste preferences and preterm babies may be capable of processing stimulations coming from their environment. The impact of sensory stimulation prior to tube feeds on clinical outcomes, such as feed tolerance, attainment of feeding skills and development of feeding preferences, as well as the impact of feeding experience and NICU environment on cortical activation, remains to be determined.

## Figures and Tables

**Figure 1 nutrients-13-00350-f001:**
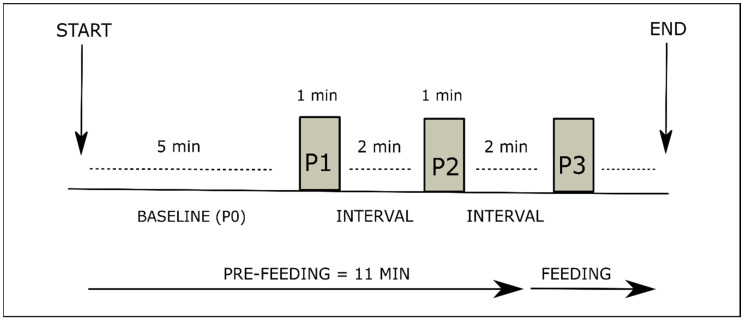
Assessment sequence. Prefeeding consisted of baseline period (P0, five minutes); period 1 (P1, one minute), corresponding to smell exposure in intervention group; period 2 (P2, one minute), corresponding to taste exposure in intervention group; and feeding period (P3, duration according to feeding volume).

**Figure 2 nutrients-13-00350-f002:**
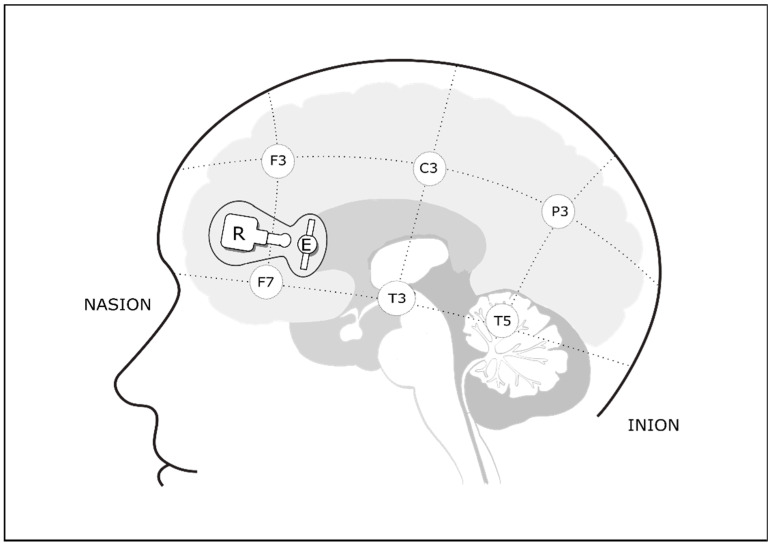
Lateral view representing optode position. Optodes were placed bilaterally with emitter (E) positioned anterior to T3/T4 and receiver (R) positioned anterior to F7/F8 according to the international 10–20 system.

**Figure 3 nutrients-13-00350-f003:**
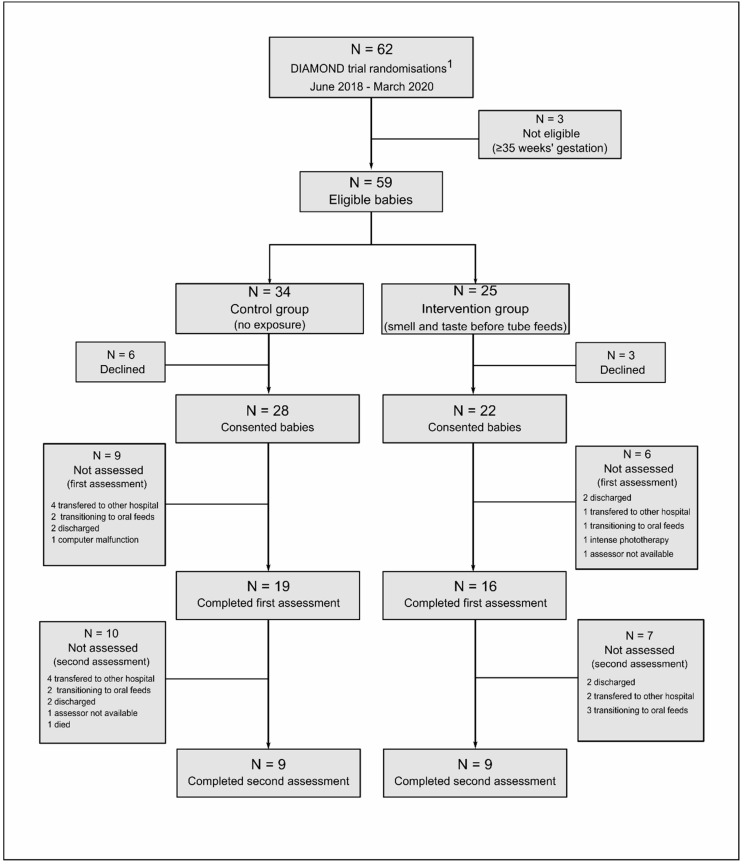
Study flowchart. ^1^ Total recruited at Auckland City Hospital during the study period.

**Figure 4 nutrients-13-00350-f004:**
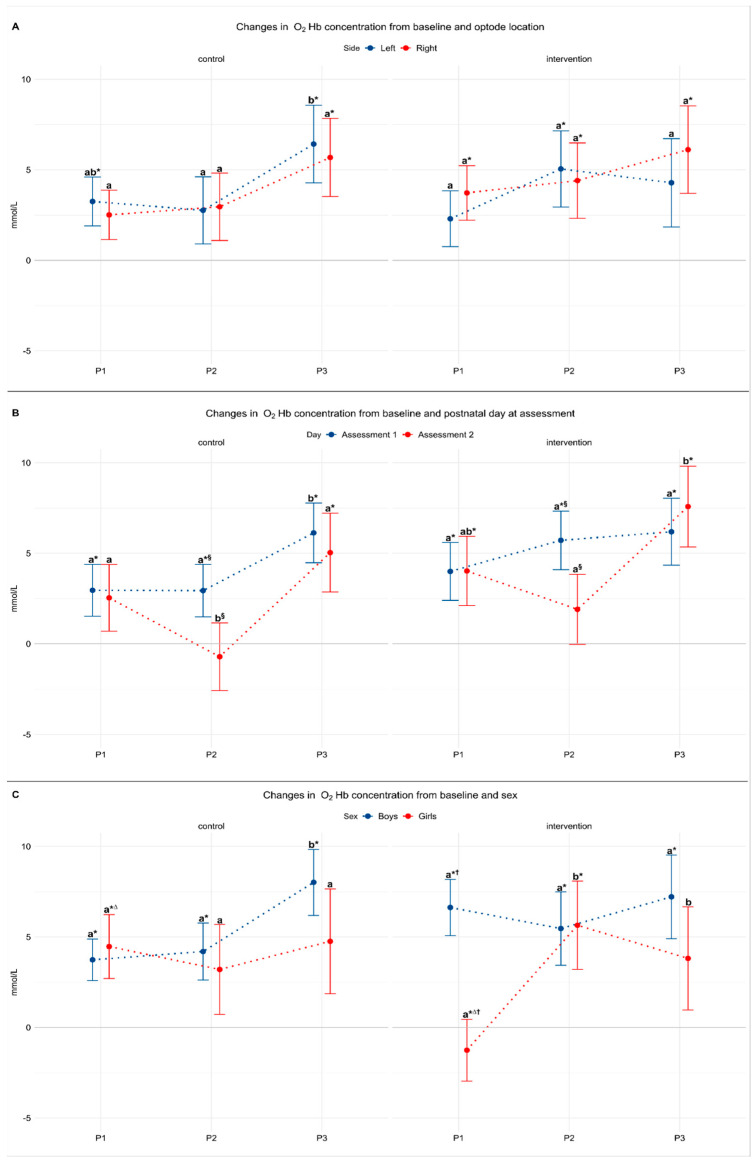
Model estimates adjusted for sex, gestation, and baseline oxygenation. Values are mean with 95% confidence intervals. (**A**) Effect of laterality (control N = 18, intervention N = 14). (**B**) Effect of postnatal day (control N = 19, intervention N = 16). (**C**) Effect of sex (control N = 18, intervention N = 14). * Significant change from baseline. ^a,b^ Different letters represent significant differences among periods. ^§^ Significant difference between assessment 1 and assessment 2. ^∆^ Significant group difference. ^†^ Significant sex difference. Statistical significance taken as *p* < 0.05 for all comparisons. P1: period 1 (smell exposure in intervention group). P2: period 2 (taste exposure in intervention group). P3: period 3 (feeding period in both groups). O_2_Hb = oxygenated hemoglobin.

**Table 1 nutrients-13-00350-t001:** Study groups’ characteristics.

	Study Group		
	Control (*n* = 19)	Intervention (*n* = 16)	*p*
Participant characteristics:			
Boys	14 (74)	10 (63)	0.59
Gestational age, weeks	33 (32–34)	33 (32–34)	0.93
Birth weight, grams	1883 (306)	1909 (393)	0.81
Birth length, cm	43.6 (2.7)	43 (3.1)	0.50
Birth head circumference, cm	30.6 (1.2)	30.4 (2.0)	0.81
Gestational age, weeks	33 (32–34)	33 (32–34)	0.93
Discharge weight, grams	2557 (299)	2411 (375)	0.16
Duration of NICU stay, days	28 (11–51)	24 (10–36)	0.30
In-hospital weight gain, g/Kg·day	10.8 (4.1)	9.0 (4.6)	0.28
Maternal age, years	35 (21–50)	32 (18–45)	0.20
Birth by Cesarean section	14 (74)	13 (81)	0.71
Antenatal corticosteroid received	18 (95)	13 (81)	0.57
At assessment 1:			
Postnatal age at assessment, days	5 (3–6)	4 (3–5)	0.15
Volume of tube feed, mL	15 (2–43)	18 (4–38)	0.61
Duration of assessment, min	20 (14–31)	21 (11–41)	0.91
Milk fed during assessment			0.61
Breastmilk only	16 (84.2)	12 (75)	
Mixed feeding ^A^	1 (5.3)	3 (18.7)	
Infant formula	2 (10.5)	1 (6.3)	
At assessment 2:			
Total participants	9 (47)	9 (56)	
Boys	8 (89)	8 (89)	
Postnatal age at assessment, days	10 (8–11)	10 (8–12)	0.86
Volume of tube feed, mL	40 (26–50)	29 (16–48)	0.30
Duration of assessment, min	27 (24–35)	26 (22–44)	0.80
Milk fed during assessment			0.38
Breastmilk only	8 (88.9)	7 (77.8)	
Mixed feeding ^A^	0	0	
Infant formula	1 (11.1)	2 (22.2)	

Data are presented as mean (SD), median (range), or *n* (%). NICU: neonatal intensive care unit. ^A^ Mixed feeding means the baby received a combination of breastmilk and formula during the feed.

**Table 2 nutrients-13-00350-t002:** Changes in oxygenated hemoglobin (mmol/L) concentrations from baseline. Mixed models estimates for three-way interaction between group, period, and (A) laterality; (B) postnatal day; (C) sex.

	Control Group (*n* = 19)			Intervention Group (*n* = 16)		
	Period			Period		
Model Estimates	P1	P2	P3	P1 (Smell)	P2 (Taste)	P3
**(A) Laterality:***F_(2,59)_* = 0.88 *p =* 0.4						
Left side	3.2 *^,a,b^ (0.5–6.0)	2.8 ^a^ (−0.9–6.5)	6.4 *^,b^ (2.1–10.7)	2.3 ^a^ (−0.8–5.8)	5.0 *^,a^ (0.8–9.2)	4.3 ^a^ (−0.6–9.2)
Right side	2.2 ^a^ (−0.2–5.3)	2.9 ^a^ (−0.8–6.7)	5.7 *^,a^ (1.4–10.0)	3.7 *^,a^ (0.7–6.7)	4.4*^,a^ (0.2–8.6)	6.1 *^,a^ (1.3–10.9)
**(B) Postnatal day:***F_(2,95)_* = 0.41 *p =* 0.7						
Assessment 1	2.9 *^,^^a^ (0.05–5.8)	2.9 *^,a,§^ (0.01–5.9)	6.1 *^,b^ (2.8–9.4)	4.0 *^,a^ (0.8–7.2)	5.7 *^,a,§^ (2.4–9.0)	6.2 *^,a^ (2.5–9.9)
Assessment 2	2.5 ^a^ (−1.1–6.2)	−0.7 ^b,§^ (−4.4–3.0)	5.0 *^,a^ (0.7–9.4)	4.0 *^,ab^ (0.2–7.8)	1.9 ^a,§^ (−2.0–5.8)	7.6 *^,b^ (3.1–12.0)
**(C) Sex:***F_(2,59)_* = 4.94 *p =* 0.01						
Boys	3.7 **^,^*^a^ (1.4–6.1)	4.2 *^,a^ (1.0–7.4)	8.0 *^,b^ (4.3–11.7)	6.6 *^,a,†^ (3.4–9.8)	5.5 *^,a^ (1.4–9.5)	7.2 *^,a^ (2.6–11.8)
Girls	4.5 *^,a,∆^ (0.8–8.1)	3.2 ^a^ (−1.8–8.2)	4.7 ^a^ (−1.0–10.6)	−1.2 ^a,∆,†^ (−4.8–2.3)	5.6 *^,b^ (0.7–10.5)	3.8 ^b^ (−1.9–9.5)

Model estimates adjusted for sex (except for C), gestation, and baseline oxygenation. Values are expressed as mean mmol/L (95% confidence interval). * Significant change from baseline. ^a,b^ Different letters represent significant differences among periods. ^§^ Significant difference between assessment 1 and assessment 2. ^∆^ Significant group difference. ^†^ Significant sex difference. Statistical significance taken as *p* < 0.05 for all comparisons.

## Data Availability

The data presented in this study are available on request from the corresponding author. The data are not publicly available as the DIAMOND trial is ongoing.

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
