# Peer review of "Cortical Oxygenation Changes during Gastric Tube Feeding in Moderate- and Late-Preterm Babies: A NIRS Study"

_nutrients, 2021, doi:10.3390/nu13020350_

Round 1
Reviewer 1 Report
The objective of the study was to evaluate the oxygenation changes in the OFC measured by NIRS in moderate-late preterm infants, following exposure to smell and taste of milk and during tube feeds. The results demonstrated that changes in cerebral oxygenation in the OFC can be detected in moderate and late preterm infants before and during tube feeding. The greatest increase in O2H concentration in the OFC occurred in response to tube feeding, suggesting that gustatory and olfactory receptors may be activated when using the feeding tube. This is an interesting conclusion and helpful findings for neonatologists that are working in NICU.
The biggest limitation of the study is the missing evaluation of the effect of type of feeding (breast milk vs infant formula vs donor breastmilk) on the oxygenation changes in the OFC. I have suggested some corrections in the following comments to reduce/eliminate this limitation.
Methods:
Line 88. How the intervention interval, time, and exposure to taste was determined (why one min of exposure)? references?
Line 124. "minimize" (not minimise)
Line 37. How the sample size was calculated to obtain sufficient statistical power? Please add this information after line 37.
Results
Line 139. Study "Population"
Line 142. Postnatal age? please specify
Table 1 misses the type of feeding, which could influence the outcomes/results of this study. Please, include the number of infants who received 1) exclusively breast milk; 2) donor milk + breast milk; 3) breast milk + infant formula; 4) donor milk; 5) infant formula.
The effect of type of feeding should be evaluated and included in Table 2.
Figure legend 4. It should specify the sample size (n value in each group)
Line 217. "among"
Author Response
We appreciate the comments and suggestions made. Please see bellow our reply:
1) The biggest limitation of the study is the missing evaluation of the effect of type of feeding (breast milk vs infant formula vs donor breastmilk) on the oxygenation changes in the OFC. I have suggested some corrections in the following comments to reduce/eliminate this limitation.
We agree that assessing the effect of type of feeding on oxygenation changes in the orbitofrontal cortex (OFC) would be of great interest. However, the majority of babies were fed breastmilk, with or without the addition of a multicomponent fortifier, during the assessments, with only 3 babies receiving formula. We have added this information was added to table 1. With an n of 3 we are unable to conduct a meaningful statistical analysis to determine the effect of type of feeding in activation of OFC. This now has been acknowledged in the discussion section (lines 288-290 and 352-354).
2) Methods Line 88. How the intervention interval, time, and exposure to taste was determined (why one min of exposure)? References?
Information regarding how assessment sequence was determined has been added to method section (lines 92-94 and 100-103).
3) Line 124. "minimize" (not minimise)
The manuscript is written with British, rather than American, English spelling so we will leave this to the type-setters, if accepted, according to the preferred style.
4) Line 37. How the sample size was calculated to obtain sufficient statistical power? Please add this information after line 37.
As this was a cohort study nested within a randomised controlled trial (DIAMOND), sample size was limited by the number of trial participants <35 weeks’ gestation at the one site with NIRS available whose parents/caregivers provided consent for NIRS assessments. Therefore, no sample size was determined a priori. This information has been added to the method section (lines 86, 88-89), and acknowledged in the discussion section (lines 349-350).
5) Line 139. Study "Population": spelling has been corrected (line 154).
6) Line 142. Postnatal age? please specify:
Information regarding postnatal age at the time of assessments has been rephrased to ensure clarity (line 157).
7) Table 1 misses the type of feeding, which could influence the outcomes/results of this study. Please, include the number of infants who received 1) exclusively breast milk; 2) donor milk + breast milk; 3) breast milk + infant formula; 4) donor milk; 5) infant formula.
This information has been added to table 1. There were no babies who received donor milk, so we have classified babies as breastmilk only, mixed feeding (breastmilk plus infant formula) and infant formula.
8) The effect of type of feeding should be evaluated and included in Table 2. Please see the response to point 1.
9) Figure legend 4. It should specify the sample size (n value in each group).
Thank you for spotting this omission. Corrected as requested (lines 212-213).
10) Line 217. "among".
Among is more common in North American spelling but both are correct; again, we are happy to leave this to the type-setters according to journal style.
Reviewer 2 Report
Dear authors,
I found the study quite interesting and easy to read and to understand. I have somme minnor comments about it.
I would include some numerical information from the results in the abstract.
The introduction is quite clear, easy to read and to understand.
How can you justify that your study is a "cohort study" if you do an intervention?
The results and the discussion clearly present the information.
Author Response
We appreciate the comments and suggestions made. See bellow our reply:
1) I would include some numerical information from the results in the abstract.
We have rephrased the abstract to allow for some numerical information within the permitted word count.
2) How can you justify that your study is a "cohort study" if you do an intervention?:
Our study is a cohort study nested within a randomised controlled trial (RCT), which combines aspects from a cohort study (the subset of babies within the RCT whose parents consented to this additional assessment) and an RCT (random allocation to be exposed or not to study intervention). This information can be found in method section line 71.
Round 2
Reviewer 1 Report
The authors answered all the requested comments.